# Forensic Facial Approximation of 5000-Year-Old Female Skull from Shell Midden in Guar Kepah, Malaysia

**Johari Yap Abdullah** [1], **Cicero Moraes** [2], **Mokhtar Saidin** [3], **Zainul Ahmad Rajion** [4], **Helmi Hadi** [5], **Shaiful Shahidan** [3,*] and **Jafri Malin Abdullah** [6,7,8,*]

1. Craniofacial Imaging Laboratory, School of Dental Sciences, Universiti Sains Malaysia Health Campus, Kota Bharu 16150, Kelantan, Malaysia
2. Ortogonline Treinamento em Desenvolvimento Profissional e Consultoria LTDA, Sinop 78557-682, MT, Brazil
3. Center for Global Archaeological Research, Universiti Sains Malaysia, Gelugor 11800, Penang, Malaysia
4. Kulliyyah of Dentistry, IIUM Kuantan Campus, Kuantan 25200, Pahang, Malaysia
5. School of Health Sciences, Universiti Sains Malaysia Health Campus, Kota Bharu 16150, Kelantan, Malaysia
6. Department of Neurosciences, School of Medical Sciences, Jalan Hospital USM, Universiti Sains Malaysia Health Campus, Kota Bharu 16150, Kelantan, Malaysia
7. Brain and Behaviour Cluster, School of Medical Sciences, Universiti Sains Malaysia Health Campus, Kota Bharu 16150, Kelantan, Malaysia
8. Department of Neurosciences & Brain Behaviour Cluster, Hospital Universiti Sains Malaysia, Universiti Sains Malaysia Health Campus, Kota Bharu 16150, Kelantan, Malaysia
* Correspondence: shaiful.shahidan@gmail.com (S.S.); brainsciences@gmail.com (J.M.A.)

**Abstract:** Forensic facial approximation was applied to a 5000-year-old female skull from a shell midden in Guar Kepah, Malaysia. The skull was scanned using a computed tomography (CT) scanner in the Radiology Department of the Hospital Universiti Sains Malaysia using a Light Speed Plus scanner with a 1 mm section thickness in spiral mode and a $512 \times 512$ matrix. The resulting images were stored in Digital Imaging and Communications in Medicine (DICOM) format. A three-dimensional (3D) model of the skull was obtained from the CT scan data using Blender's 3D modelling and animation software. After the skull was reconstructed, it was placed on the Frankfurt plane, and soft tissue thickness markers were placed based on 34 Malay CT scan data of the nose and lips. The technique based on facial approximation by data extracted from facial measurements of living individuals showed greater anatomical coherence when combined with anatomical deformation. The facial approximation in this study will pave the way towards understanding face prediction based on skull structures, soft tissue prediction rules, and soft tissue thickness descriptors.

**Keywords:** forensic facial approximation; computed tomography; face prediction; 3D modelling; skull reconstruction; facial reconstruction

## 1. Introduction

Forensic facial reconstruction (FFR) aims to identify a person by reconstructing the face from the skull. This method combines anatomical, anthropological, artistic, and graphic principles to reconstruct living appearance based on skull morphology for recognition or identification. FFR has been used since the end of the 19th century; the technique is still debated [1] due to the difficulty in defining art and science in such an approach and the precision of the real anatomical regions in relation to the reconstructed one.

Guar Kepah is a Pulau Pinang archaeological site. It consists of three shell middens labelled as Guar Kepah A, B, and C. The name Guar means hillock, while Kepah refers to the species of bivalve (*Meretrix meretrix*) commonly found on the coast and estuary of Sungai Muda. Guar Kepah is the first archaeological site excavated by the British in Peninsular Malaysia [2]. Archaeological research in Guar Kepah started in the second half of the 19th century. In 1860, Earl [3] surveyed a canal that had become a dispute among the Malay paddy-planters near Sungai Muda in Province Wellesley (now Seberang Perai).

He observed a small Chinese settlement where lime was burned from the "cockle-shells" mound. The dome-shaped mound was approximately eighteen feet (5.5 m) in height and "nearly two hundred paces" (152.4 m) in circumference. A quarter of this mound had been dug off. According to Earl, there was a clear indication that the mound was formed by human agency, and the shells were discarded after the fish had been removed; however, the locals believed it was formed by nature [3]. Earl also collected human remains and stone tools discovered by the workers and shipped them to England for further analysis and observation.

For the next 150 years, researchers studied the human remains of Guar Kepah [4–7], the location of the sites [8], and the Hoabinhian stone tools and conducted comparative synthesis with shell middens in Sumatera, Indonesia [9]. Callenfels, Evans, and Tweedie conducted the first archaeological excavation in all three sites in June 1934. Despite the fact that all of the Guar Kepah shell middens were severely disturbed due to the lime kiln activities, the excavations managed to discover stone tools, pottery, human remains, faunal remain, beads, and various species of shells such as *Meretrix meretrix*, *Arca granosa*, *Melongena pugilina*, Ostrea (Rivularis and *Turitella attenuata*) [10].

In 2010, a team of archaeologists from the Centre of Global Archaeological Research (CGAR), Universiti Sains Malaysia unearthed stone tools, pottery, ornaments, and faunal remains in the second archaeological excavation since 1934. The excavation was conducted in Guar Kepah B. No human remains were found, but the findings showed Neolithic culture in a shell-midden setting. However, a subsequent excavation on the same site by CGAR was performed in 2017, and skeletal remains were discovered and labelled as GKph2017. The skeleton was found with both hands flexed to the chest, and it is believed that the body was laid in a semi-flexed position. All upper body parts are almost complete, except for some missing bones such as the carpal, metacarpal, and upper right lateral incisor. The skeleton was laid in the north-west, south-east orientation, lying on the right side with the face facing south-west. Physical examination of the skeleton was conducted when the skeleton was exposed. A proper burial ceremony can be observed based on the stratigraphy and the placement of burial goods (stone tools and pottery) around the skeleton [11].

Due to missing lower abdominal parts, the gender of GKph2017 was determined by non-metric cranial morphology analysis [12] and Walker's score system on specific landmarks in the cranium [13] due to the missing lower abdominal parts. The same sexing scoring method was used for consistency and to create a comparable result with the previous research on Guar Kepah skeletons, as well as to reflect the original article and materials when scoring features and utilising methods [14]. Based on Larnach and Freedman [12], a sexing score of 8/18 (0.4) was given to GKph2017, which signifies a female gender. These scores (including Walker's) are consistent with other female skeletons from Guar Kepah analysed by Bulbeck [7]. Tooth wear and dental development were based on Ubelaker [15], Gustafson and Koch [16], and Anderson et al. [17]. The age of death was also determined via cranial suture closure by Meindl and Lovejoy [18]. The presence of the third molar in both the lower left and right quadrant (tooth no. 38 and 48; FDI notation) suggests that the skeleton is at least 19 years old. A clear, relatively heavily worn sign on the cusp of teeth 37 and 38 could suggest a hard, gritty diet. A tooth wear score by Lovejoy [19] marks the age at death of around 35–45 years old, while a cranial suture closure averages the age of death at 39.4–41.1 years old. Therefore, it is concluded that the age of death for GKph2017 is around 40 years old. The stature estimation is based on the right and left humeri, which are still intact. The regression formula uses Sjøvold [20], a weighted line of organic correlation applicable to a wide range of populations, independent of sex. The estimated stature for GKph2017 was 150.44 ± 4.89 cm. Chronometric dating shows GKph2017 was buried around 5700 years ago.

## 2. Materials and Methods

The GKph2017 skull was scanned using a computed tomography (CT) scanner in the Department of Radiology at Hospital Universiti Sains Malaysia (Kota Bharu, Malaysia)

with a 1 mm slice thickness in spiral mode and a 512 × 512 matrix (Figure 1). A 200 mA tube current at 120 KVP was used to perform the scanning. Digital Imaging and Communications in Medicine (DICOM) format were utilised to store the generated pictures. The CT scan data were used to generate a three-dimensional (3D) model of the skull using Blender's 3D modelling and animation tools (blender.org, accessed on 14 May 2022). The forensic add-on for Blender (ForensicOnBlender) is a sequence of commands that enables the user to conduct digital forensic facial reconstruction within Blender 3D software. It is a solution built on open-source software that is freely accessible for the three most common operating systems on the market: Windows, Linux, and Mac OSX [21].

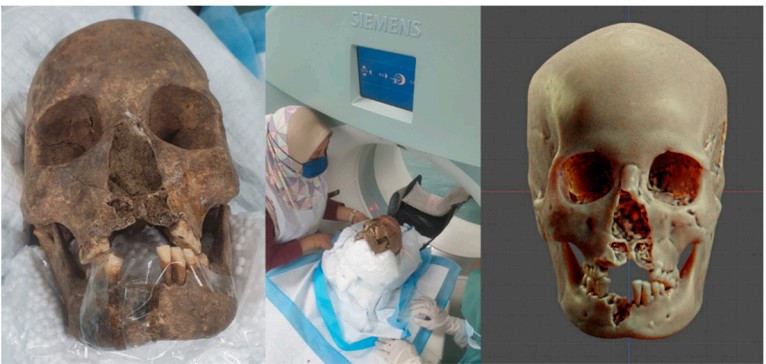

**Figure 1.** Scanning of the GKph2017 skull and the resulting 3D model of the skull.

After the skull was reconstructed, it was positioned on the Frankfurt plane and soft tissue thickness markers were placed based on CT scan data of the nose and lips from 34 Malay individuals. In addition to the measurements presented in the book chapter by Moraes et al. [22], the height (Z-axis) of the ears and the eye openness (X-axis) were measured based on Malay data [23].

### 2.1. Lateral Nasal Projection (Y-Axis)

Figure 2 depicts the initial measurements according to the study published by Moraes et al. [24] with *n* = 34. The mean aperture is 28.5 mm, and the standard deviation (SD) is 2.6. The mean for the Base is 30 mm, with a SD of 3.1 mm. The mean Aperture–Base Angle is 54.4o with an SD of 5.4. The mean distance between the front nasal spine (Gerasimov technique) and the columella is 6.1 mm, with an SD of 1.7 mm.

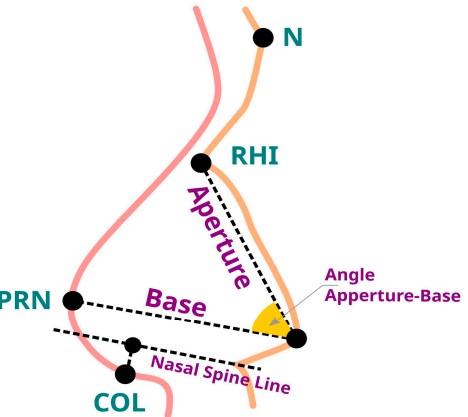

**Figure 2.** The study used the distance from the rhinion to the lateral border of the nose, called Aperture, and the distance between the pronasale and the lateral border, called the Base. Another distance measured was between the projected line from the lower nasal crest to the columella. With them, it was possible to stipulate the size of the Base from Aperture. The end of the Base is the pronasale, and the columella receives an average increment from the measurements made.

## 2.2. Facial Measurements

The facial measurements presented here are consistent with Moraes et al.'s [22] research. These measurements highlight the significance of key spaces that could be used as a basis for the projection, such as the distance between the frontomalar orbital points (fmo–fmo), which is useful for a variety of projections, such as the positioning of the eyeballs, the frontal dimension of the nose (X-axis), and the lips. These points are in accordance with the description proposed by Caple and Stephan [25]. The mean fmo–fmo distance is 96.8, with a SD of 4.1.

In relation to the fmo–fmo points, two other distances proved effective for the projection of the lips and the frontal portion of the nose. As illustrated in Figure 3, these were obtained from the sum of the distances between the infraorbital and mental foramina.

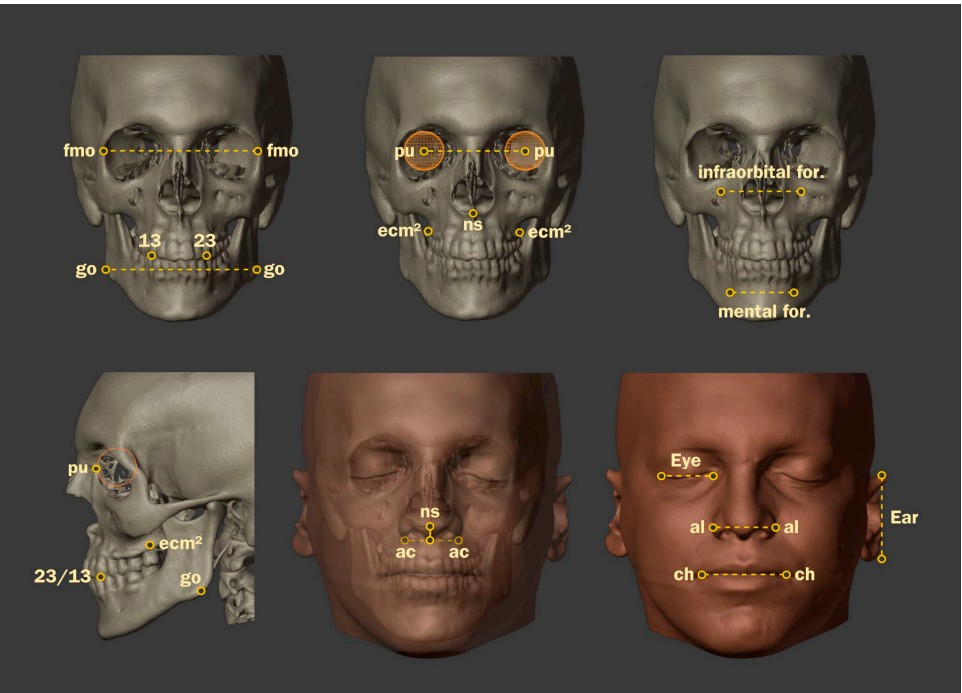

**Figure 3.** Important face measurements.

Since not all the 34 CT scans contained a complete face, this study's data were smaller than those of previous studies. The mean distance ($n = 30$) between infraorbital foramina is 53.7 mm, with a standard variation of 3.8 mm. The average distance between mental foramina ($n = 10$) is 47.9 mm, with a SD of 2.7%.

## 2.3. Three-Dimensional Positioning of the Eyeball

The location of the eyeball within the orbit was divided into three steps corresponding to the X, Z, and Y axes due to the anatomical intricacy of the region. The order is based on 3D positioning procedures. The mean distance between the eyeballs (pu–pu) is 64.4 mm with a SD of 3.4 mm. Distance between the fmo and orbit centre (orb side-centre) ($n = 34$) is 16.2 with a SD of 1. Comparing pu–pu to fmo–gmo distances, the mean ($n = 34$) is 66.6 percent with a SD of 1.9.

Regarding the Z-axis (orb Z), a line was drawn from the intersection of the lateral limit, representing the previously approached space, to the centre of the eyeball. The mean is 15.4 mm, with an SD of 1.3 mm. For Y-axis positioning (orb Y), a mean distance was determined between a tangent to the infraorbital margin and the pupil (pu). The mean distance ($n = 34$) is 6.3 mm, with an SD of 3 mm. To determine the location of the eyeball, the distance between the canine and pupil ((13–23)–pu) was determined. The mean ($n = 18$) is 1.9 mm in front of the canines, with an SD of 4.3 mm.

### 2.4. Mouth Measurement and Projection (ch-ch)

The distance between the chellions (ch–ch) is one of the most important parts of the frontal projection of the face. Although this material does not explore the heights of the upper and lower lips, the data below are useful for approximating the face's real structure.

In this sample (*n* = 19), the mean ch–ch distance is 49.3 mm, with a 3.5 SD. When calculating half of the sum of the infraorbital and mental foramina (AV foramina) versus the average distance from the lips, the results were very close to each other. In this case, the ch–ch (*n* = 19) is 49.3 mm versus 51 mm of the AV foramen (*n* = 9). The horizontal line of the mouth can also be projected using ch–ch and fmo–fmo (percent ch vs. fmo). The ch–ch space is 50.7% of the fmo–fmo space (*n* = 19), with an SD of 3.3.

### 2.5. Measurements and Frontal Positioning of the Nose

Similar to the eyes, the nose is a complicated structure whose anatomical alignment takes multiple stages along distinct axes. Previously, the projection of the nose profile was discussed. This portion measured the nose's frontal location. The average distance between wings (al–al) (*n* = 34) is 53.7 mm, with an SD of 3.8 mm. The distance between the canines (Teeth 13–23) is a significant indicator of nose projection (*n* = 19, mean 34.4 mm, SD 2.2). Next, the percentage representing the al-la distance with respect to the total of the lengths between the infraorbital foramina and distance between the canines was increased (percent al vs. InfrCan). The mean (*n* = 17) is 48%, with an SD of 4.3.

Another projection-based approach compares the al–al versus fmo–fmo distance, which is useful when foramen or canines are missing. The mean is 43.3% and the SD is 3.8. The data above place the wings on the X-axis, and after lateral projection, on the Y-axis. The Z-axis is not present. Distance from the nasospinate point to nasal wings (ns–ac) was measured. The mean (*n* = 34) is 5 mm, with an SD of 2.3.

### 2.6. Eyelid Opening Measurement (Eye)

Eyelid openings (Eye) were measured by monitoring frontally drawn lines for other measures that matched with eyelid limits (Figure 4), Specifically, the alar (al–al) and frontomalar (fmo–fmo) vertical lines. The mean of the eyelid opening (*n* = 34) is 29 mm, with a 1.7 SD.

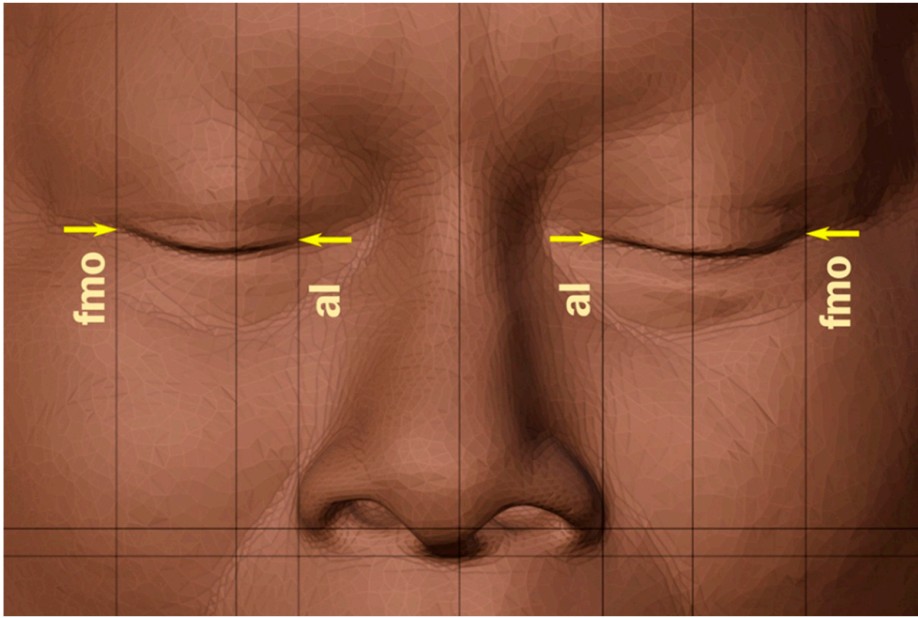

**Figure 4.** The fmo–fmo and al–al lines show compatibility with the eyelid extremities in a series of studies. However, the projection was inferior to that based on 30% of the fmo–fmo distance.

Using the produced data, the authors calculated the projection of the eyelid opening by subtracting the fmo–fmo distance from the al–al distance and dividing the result by two (((fmo–fmo) − (al–al))/2). If, on the one hand, a series of results were consistent with the real measurement, a significant number of others were not. Out of curiosity, the authors plotted the average percentage of the eyelid opening space against the fmo–fmo distance, as this distance was compatible with a number of others. Unexpectedly, this projection performed better than the al–al and fmo–fmo tracings, and even better than the regional average, with the added benefit of not requiring the reconstruction of a soft tissue structure (al-al). Comparing the eyelid opening distance (Eye AV) to the fmo–fmo distance, the mean (*n* = 34) and SD are 30% and 1.6, respectively.

### 2.7. Ear Measurement

It is impossible to reconstruct the ears since the skull has only the exterior acoustic meatus. The ear is typically a secondary component in facial approximation since observers concentrate on the face's frontal region. The authors examined the ears initially to establish an average Z-axis (height) dimension but later discovered an unanticipated link with the fmo–fmo distance (fmo* 60.7 percent). Because the ear and fmo–fmo line are on opposite axes, this was unexpected. The mean ear height (*n* = 29) is 58.3 mm with an SD of 5.1%. The ear height vs. fmo–fmo distance (*n* = 29) averaged 60.7% with an SD of 5.8%.

### 2.8. Other Measurements

Due to their compatibility with other sections of the skull, two additional measurements were listed: the distance between the cervical bone crests of molar teeth 16 and 26 (emc2–emc2) and the distance between the gonion's frontal extremities (go–go). The average distance between emc2 and emc2 is 65.3 mm, with an SD of 3.6 mm. The distance between the go–go spots is 97.4 mm, with an SD of 3.5%.

### 2.9. The Forensic Facial Approach

The authors prefer to use the term forensic facial approximation [26] rather than forensic facial reconstruction because the elements used to model the face are the result of projections from population data, with their margins of error, and no technique guarantees an accurate reconstruction of an individual's face. Therefore, the first term corresponds better with the proposed work.

All work was performed using the OrtogOnBlender add-on [27] and its sub-module, ForensicOnBlender, in space. These tools enable the Blender software to perform tasks not available in the native version, such as the digitisation of 3D objects by photography, 3D tomography reconstruction, and complex Boolean calculations for simulating osteotomies.

Figure 5A illustrates the 3D reconstruction of GKph17 tomography with a 200 threshold [28]. A distorted 3D mesh from a virtual donor was used to reconstruct the lost maxilla and mandible (Figure 5B). The recovered skull was positioned in accordance with the Frankfurt plane. The distribution of soft tissue thickness markers was based on a study involving Chinese adults whose anatomical locations were assessed using ultrasonography [29]. The selection adhered to the requirement of a living Asian population; thus, the approximation more closely matched the anatomy, which is not the case with thickness tables based on deceased individuals, where rigour mortis sets in.

The next stage was to profile the nose using a combination of procedures, but primarily the one already discussed, which was based on Malay anatomical components, particularly with the addition of a 5 percent projection of the Nose Base instead of the 12 percent projection utilised for Brazilians. The step-by-step approach can be viewed in Moraes et al. [24] for more information. An orthographic facial profile was constructed using soft tissue markers and nasal projection (Figure 5C).

The focus then shifted to frontal projections, including eyeball placement, nasal wing height, and mouth size (Figure 5D). These projections were provided with step-by-step instructions by Moraes et al. [22]. This study increased prior eye and ear size estimations

(Figure 5D,I). Despite the fact that markers and projections provide adequate approximation references, there are a number of locations that are not aided by that approach and others that are susceptible to artistic subjectivity in the face of real anatomy. In light of this, the authors imported two CT scans of Malay individuals for use in the anatomical deformation technique (Figure 5E). This technology permits the deformation and reconstruction of the donor's skull mesh and soft tissue until the meshes conform to the approximated individual. This technique is valuable for the study of fossilised extinct creatures [30] and humans [31]. It can be used to recreate extinct species using modern equivalents.

Both CT scans in this investigation revealed comparable nasal deformity (Figure 5F,G). The creators used a structured file of a young European woman who died 400 years ago in Tabor, Czech Republic for the final face (https://www.irozhlas.cz/veda-technologie/historie/tricetileta-valky-tabor-hustiske-muzeum1812151239pj, accessed on 25 May 2022). A series of structural adjustments allowed the mesh to be modified (deformed) to conform to projections such as lines and soft tissue thickness indicators (Figure 5H,I) while maintaining the anatomical deformation's fine features. Additionally, the skin texture was altered to match the fossil ancestry. The hairs were modified with a configuration file (Figure 5J).

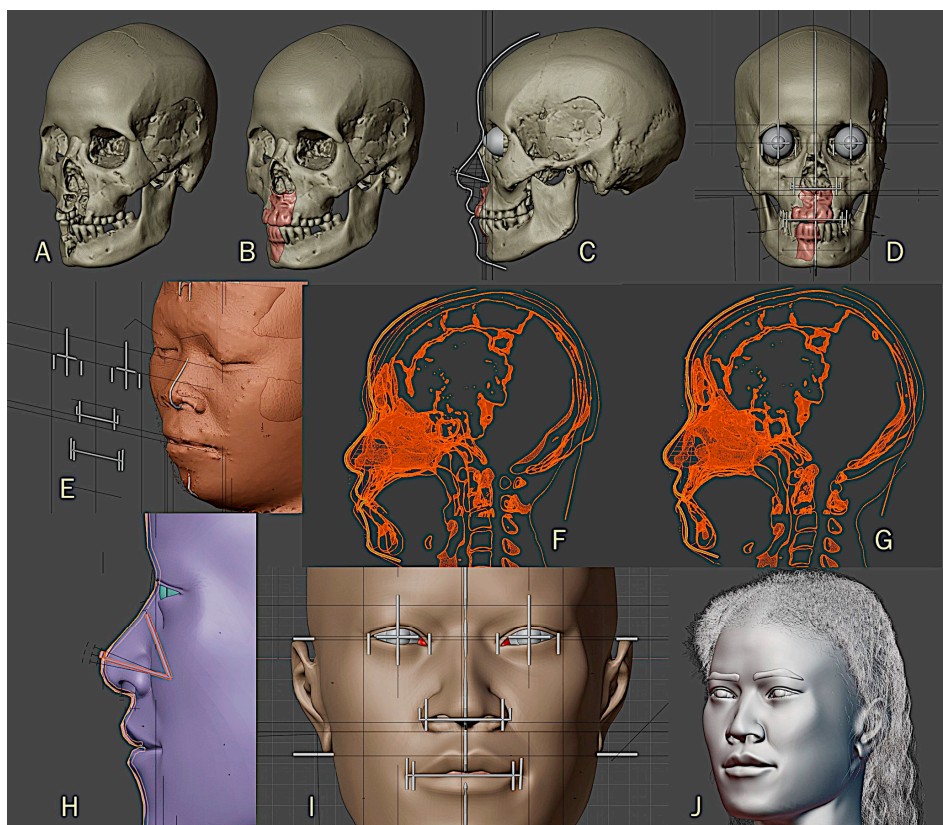

**Figure 5.** The steps of forensic facial approximation. (**A**) The reconstructed skull was incomplete; thus it was filled with a 3D mesh from a virtual donour (**B**). A profile line of the face was drawn (**C**) and the frontal projections was generated (**D**). Anatomical deformation technique was performed to approximate the face (**E**). Comparable nasal deformity was observed (**F,G**). A series of structural modifications was performed to adapt with the projections (**H,I**). The hairs were added to complete the facial approximation (**J**).

## 3. Results and Discussion

Final images were generated using Blender 3D's Cycles renderer. Four shots from different angles (Figure 6) show the volumetric anatomy of the face.

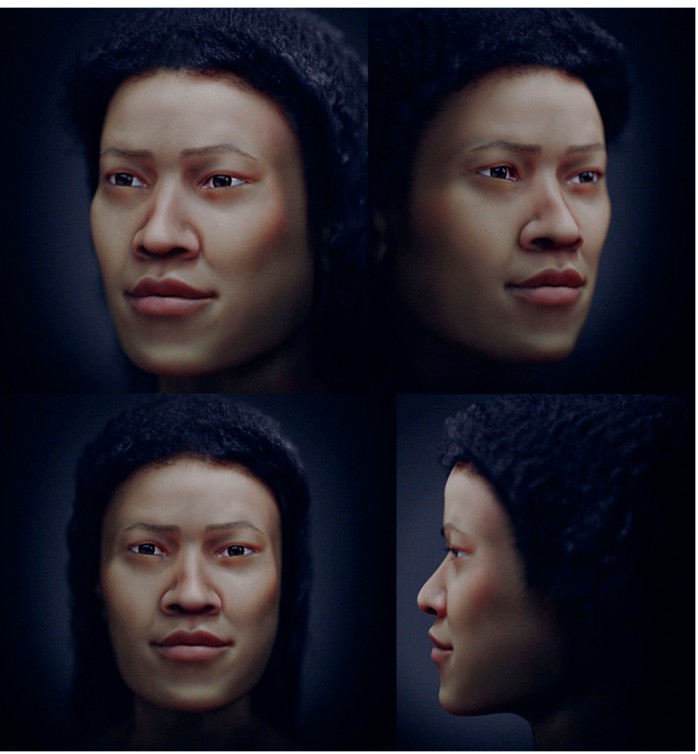

**Figure 6.** The final image of GKph2017.

The authors measured the intracranial volume, which is 1330 mL, comparable to modern humans, which averages 1328 mL [32,33]. The technique based on facial approximation by data extracted from facial measurements of living individuals showed greater anatomical coherence when combined with anatomical deformation.

Guar Kepah is recorded as a site where both Australomelanesoid and Mongoloid features could be observed within the human remains discovered [7,34,35]. Jacob [6] mentioned that some molar features resemble Australomelanesoid and some are intermediate between the Mongoloids and the Melanesians. These irregular features, including a detailed analysis of 37 individuals of Guar Kepah by Jacob, concluded that the Guar Kepah population is a "mixture of Mongoloid and Australomelanesian races".

Bulbeck [7] inventoried Guar Kepah human remains stored at the Naturalis Biodiversity Center in Leiden, The Netherlands, and concluded that there are 41 individuals from all three shell middens. GKph2017 is the only known Guar Kepah skeleton in Malaysia, though access to the material stored in Leiden is currently limited. The analysis of the human remains shows that the ancient people of Guar Kepah belong to the Melanesoid group, with similar affinities to the modern inhabitants of New Guinea, Australia, and Melanesia [9]. Bulbeck [7] also suggested that the cranial analysis is compatible with the "transitional Neolithic" period.

Wilkinson [36] compiled a set of variables to determine racial origins. The Mongoloid features a rounded orbital margin, medium width nasal aperture, moderate prognathism, absent brow ridge, straight nasal, and wide facial breadth [37,38]. The face is flatter, broader, and squarer with prominent cheekbones and a straight profile [39]. However, the Australomelanesoid displays a squarer orbital margin, a large and elongated mastoid process, and a clear temporal line in the parietal region. Most features from both categories could be observed in the GKph2017 skeleton. The facial reconstruction of GKph2017 mapped and enhanced those features, further supporting Jacob's [6] argument that interbreeding in the Guar Kepah population "had not occurred for a long time as the racial characteristics are not homogenous".

To date, no facial forensic approximation has been made on any human remains from Guar Kepah. This process could help identify any disease or trauma that affected an

individual during their lifetime [40]. The reconstruction could also suggest an alternative method of discerning archaeological skeletal remains' biological and racial identities and calls for a reassessment of the current affinities associated with Guar Kepah skeletons. Part of the process and technique indirectly promotes digital documentation in the conservation of human remains in Malaysia and enhances the craniometric and morphological analysis.

## 4. Conclusions

The facial approximation of a 5000-year-old female skull in this study will pave the way towards understanding face prediction based on skull structures, soft tissue prediction rules, and soft tissue thickness descriptors. Despite a small sample in relation to the population, it established promising predictive references for orbital positioning, mouth size, and nose characteristics. Knowing that facial reconstruction is a technique based on approximation of facial characteristics based on population averages, the high correlation index between the estimates encourages putting the proposed technique into practice, favouring further research on the method and anticipating benefits for facial reconstruction work.

**Author Contributions:** Conceptualisation, J.Y.A., J.M.A. and M.S.; methodology, C.M., J.Y.A. and H.H.; investigation, S.S. and M.S.; resources, M.S.; data curation, S.S.; writing—original draft preparation, J.Y.A., C.M., S.S. and H.H.; writing—review and editing, J.Y.A., S.S., Z.A.R. and M.S.; visualisation, C.M.; supervision, J.M.A., M.S. and Z.A.R.; project administration, S.S.; funding acquisition, M.S. All authors have read and agreed to the published version of the manuscript.

**Funding:** The study was made possible with the funding from the Chief Minister Incorporated, Penang State Government through Universiti Sains Malaysia (grant no. 304/PARKEO/650894/K104).

**Institutional Review Board Statement:** Not applicable.

**Informed Consent Statement:** Not applicable.

**Data Availability Statement:** Data are available upon request.

**Acknowledgments:** Thanks to the Department of Radiology, Hospital Universiti Sains Malaysia, for assisting the scanning of the skull, and thanks to Davi Sandes Sobral for providing the tomography used in the didactic image.

**Conflicts of Interest:** The authors declare no conflict of interest.

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
