# Peer review of "Forensic Facial Approximation of 5000-Year-Old Female Skull from Shell Midden in Guar Kepah, Malaysia"

_applsci, doi:10.3390/app12157871_

Round 1
Reviewer 1 Report
The article has the merit to present the first forensic facial approximation of the skull of a woman from the archaeological site of Guar Kepah. No FFA has been made on any human remains from there and this could be use for comparative studies. Good use of Ubelaker that provides good performance in age-at-death estimates. I would consider recent publications on similar topics to support your findings.
Line 71. Please add a bit of information about the archaeological mission in 2010, institutions involved to give context.
Line 73. As above for 2017.
Line 81. reference for burial good retrieved around the skeleton.
Line 84-5. "Based on Larnach and Freedman (11)". There are more recent discussions and methods around sex determination.
See for instance "Sex estimation of the human skeleton history, methods, and emerging", Alexandra Klales, 2020.
Can these confirm the results based on a 1964 study? I suggest to use some for comparison to support your evidence.
Especially because other studies demonstrated lower accuracy of sex classification when using Walker Cranial Nonmetric Method. See here for example: https://onlinelibrary.wiley.com/doi/10.1111/1556-4029.13013
Line 87. Bulbeck should be (7) rather than (4).
Line 101. Bulbeck should be (7) rather than (4).
Line 107. Bulbeck should be (7) rather than (4).
Line 129. Frankfort Horizontal Plane
Figure 2. A legend would help. Or refer to Nasion, Rhinion, Pronasale.... in the section above the image when discussing it.
Line 233. Unclear how it performed better.
Line 261-262. Indeed. In particular, for facial soft tissue thicknesses measurement errors appear to be large and have not been sufficiently documented. https://doi.org/10.1016/j.forsciint.2019.109965
As the article mentions challenges with ear approximation, Nasal representation is a challenge as well and documented in recent publications.
Line 282-283. Vague. Any technique not previously stated?
Line 338. 5,000 not 5000 for consistency.
Author Response
|
Line |
Comments |
Corrections |
|
71 |
Please add a bit of information about the archaeological mission in 2010, institutions involved to give context. |
In 2010, a team of archaeologists from the Centre of Global Archaeological Research (CGAR), Universiti Sains Malaysia, unearthed stone tools, pottery, ornaments, and faunal remains in the second archaeological excavation after 1934. The excavation was conducted in Guar Kepah B. |
|
73 |
As above for 2017. |
No human remains were found, but the findings showed Neolithic culture in a shell-midden setting. However, a subsequent excavation on the same site by CGAR was performed in 2017, and the skeleton remains were discovered and labelled as GKph2017. |
|
81 |
reference for burial good retrieved around the skeleton.
|
A proper burial ceremony can be observed based on the stratigraphy and the placement of burial goods (stone tools and pottery) around the skeleton (Shahidan et al., 2018).[11] |
|
84-85 |
"Based on Larnach and Freedman (11)". There are more recent discussions and methods around sex determination. See for instance "Sex estimation of the human skeleton history, methods, and emerging", Alexandra Klales, 2020. Can these confirm the results based on a 1964 study? I suggest to use some for comparison to support your evidence. Especially because other studies demonstrated lower accuracy of sex classification when using Walker Cranial Nonmetric Method. See here for example: https://onlinelibrary.wiley.com/doi/10.1111/1556-4029.13013
|
Added this line:
The same sexing scoring method was used for consistency and to create a comparable result with the previous research on Guar Kepah skeletons, and to reflect on the original article and materials when scoring features and utilising methods [14]. |
|
87 |
Bulbeck should be (7) rather than (4). |
Corrected to [7] |
|
101 |
Bulbeck should be (7) rather than (4). |
Corrected to [7] |
|
107 |
Bulbeck should be (7) rather than (4). |
Corrected to [7] |
|
129 |
Frankfort Horizontal Plane
|
Added this line to the Figure2: The study used the distances from the rhinion or the lateral border of the nose, called Apperture, also the distance between the pronasale and the lateral border, called Base. Another distance to start was that between the projected line from the lower nasal crest to the columella. With them it was possible to stipulate the size of the Base from Apperture. The end of the Base is the pronasale and the columella receives an average increment from the measurements made. |
|
233 |
Unclear how it performed better.
|
Performed better based on our previous study. |
|
261-262 |
Indeed. In particular, for facial soft tissue thicknesses measurement errors appear to be large and have not been sufficiently documented. https://doi.org/10.1016/j.forsciint.2019.109965
As the article mentions challenges with ear approximation, Nasal representation is a challenge as well and documented in recent publications.
|
Agree with reviewer |
|
282-283 |
Vague. Any technique not previously stated? |
This article explained the step-by-step of the reconstruction process. The cited references also provided text and video explanation. |
|
338 |
5,000 not 5000 for consistency. |
Corrected to 5,000 |
Reviewer 2 Report
Dear Authors,
Nice and well done work. The presentations was very good and understandably. One single required – you should have some discussions too, so, I recommend you to write results and discussions (together) and you can take the last paragraph from introduction and put it in the discussions part, and also you can add another 1-2 paragraph for the discussions part.
Author Response
Dear Authors,
Nice and well done work. The presentations was very good and understandably. One single required – you should have some discussions too, so, I recommend you to write results and discussions (together) and you can take the last paragraph from introduction and put it in the discussions part, and also you can add another 1-2 paragraph for the discussions part.
Answer:
Manuscript updated.
Results and discussion was put together and the last paragraph of introduction was moved to Discussion. Paragraphs added in discussion part.
Reviewer 3 Report
1. The common size of facial structures should be provided first for the kind of persons. Do diferent kinds of nationals in different environments depend on your measurements?
2. From the various measurements, the 3D model reconstruction on the screen should be provided based the used software.
3. Some grammars or typos should be checked again.
Author Response
1. The common size of facial structures should be provided first for the kind of persons. Do diferent kinds of nationals in different environments depend on your measurements?
Answer:
We can only compare with the skull from the similar time period, even then it would not be from the same place.
The comparison is a bit pointless. Therefore, we did the comparison in relation to contemporary human.
2. From the various measurements, the 3D model reconstruction on the screen should be provided based the used software.
Answer:
The 3D reconstruction was provided step-by-step in the manuscript. The current methods are based on previously published methods as cited in the manuscript.
3. Some grammars or typos should be checked again.
Answer:
Grammars and typos were corrected as suggested.